# Interventions Provided by Physiotherapists to Prevent Complications After Major Gastrointestinal Cancer Surgery: A Systematic Review and Meta-Analysis

**DOI:** 10.3390/cancers17040676

**Published:** 2025-02-17

**Authors:** Sarah White, Sarine Mani, Romany Martin, Julie Reeve, Jamie L. Waterland, Kimberley J. Haines, Ianthe Boden

**Affiliations:** 1School of Health Sciences, University of Tasmania, Launceston, TAS 7250, Australia; romany.martin@utas.edu.au (R.M.); ianthe.boden@utas.edu.au (I.B.); 2School of Allied Health, Exercise and Sports Sciences, Charles Sturt University, Albury, NSW 2640, Australia; 3Department of Physiotherapy, Launceston General Hospital, Launceston, TAS 7250, Australia; sarine.mani@ths.tas.gov.au; 4School of Clinical Sciences, Faculty of Health and Environmental Studies, AUT University, Auckland 1010, New Zealand; julie.reeve@aut.ac.nz; 5Department of Physiotherapy, The University of Melbourne, Parkville, VIC 3052, Australia; jamie.waterland@petermac.org; 6Department of Health Services Research, Peter MacCallum Cancer Centre, Parkville, VIC 3052, Australia; 7Department of Critical Care, Melbourne Medical School, The University of Melbourne, Parkville, VIC 3052, Australia; kimberley.haines@wh.org.au; 8Department of Physiotherapy, Western Health, St Albans, VIC 3021, Australia

**Keywords:** gastrointestinal cancer, gastrointestinal surgery, postoperative complications, postoperative pulmonary complications, pneumonia, physiotherapy, preoperative, perioperative, postoperative, length of stay

## Abstract

Complications after major surgery for gastrointestinal cancers are common. Physiotherapists are frequently employed by hospitals to treat patients, with interventions aimed at minimising complications and improving recovery after surgery. At present, the most effective physiotherapy interventions for this patient population are unknown. It is unclear if providing physiotherapy confers additional benefit compared to no treatment. Additionally, it is unknown if the timing of physiotherapy interventions, preoperatively, perioperatively or postoperatively, impacts outcomes. Currently, no systematic review has specifically examined the efficacy of physiotherapy in the gastrointestinal cancer surgery population. This systematic review will evaluate and synthesise the evidence, examining the effects of perioperative physiotherapy interventions delivered with a prophylactic intent on postoperative outcomes.

## 1. Introduction

Surgery is often necessary for the curative management of colorectal, upper gastrointestinal and hepatobiliary cancers. Major gastrointestinal surgery does not come without risk, with up to 50% of patients suffering a postoperative complication [1,2], which are associated with poorer outcomes, prolonged recovery for patients [3] and significant financial ramifications for the health service [4]. Understandably, the implementation of strategies to prevent postoperative complications are strongly advised [1].

Physiotherapists are commonly employed by hospitals in high-income countries to preoperatively assist patients to prepare for surgery [5] and postoperatively to aid a patient’s physical recovery and to prevent postoperative complications [6], with a particular focus on postoperative pulmonary complications (PPC) [6]. The most effective physiotherapeutic methods for preventing complications after abdominal surgery remain uncertain [7,8], predominantly due to a lack of clarity on whether additional physiotherapy interventions provide further benefit to early postoperative mobilisation alone [8]. Additionally, effect differences according to the timing of physiotherapy interventions; given either preoperatively, postoperatively, or perioperatively is not clearly understood.

The most recent systematic reviews specifically exploring physiotherapy interventions in the abdominal surgery population were published in 2006 [9,10]. Since 2006, new randomised controlled trials (RCTs) have been published that would not have been considered in these reviews. Importantly, neither of these systematic reviews were specific to the gastrointestinal cancer surgery population. Considering the large worldwide volume of gastrointestinal cancer surgery and the ubiquitous provision of physiotherapy services to these patients in high-income countries, it is essential that the evidence specific to physiotherapy in the gastrointestinal cancer surgery population is explored. Additionally, performing an updated systematic review will allow the adoption of modern statistical analysis techniques that were not available for earlier reviews.

This systematic review will evaluate and synthesise the evidence for perioperative physiotherapy interventions delivered with a prophylactic intent to effect postoperative outcomes when compared to a no-treatment control, or usual care of early mobilisation alone, in adults undergoing gastrointestinal surgery for cancer. The primary outcome will be PPCs with a sub-group analysis of effect due to timing of physiotherapy interventions. Impacts to other outcomes including medical and surgical complications, acute length of stay, functional outcomes, all-cause mortality and adverse events attributable to physiotherapy will also be explored.

## 2. Materials and Methods

### 2.1. Protocol and Registration

The protocol for this systematic review and meta-analysis was prepared in line with the Preferred Reporting Items for Systematic Review Protocols (PRISMA 2020) [11] and prospectively registered with the International Prospective Register of Systematic Reviews (PROPSERO CRD42024578131). There were no deviations to the published protocol.

### 2.2. Study Eligibility Criteria

Studies eligible for inclusion needed to have met the following criteria (Table 1).

#### 2.2.1. Design

Randomised controlled, pseudo-randomised and cluster randomised trials.

#### 2.2.2. Participants

Due to the specific focus of this review on the major gastrointestinal cancer surgery population, studies were only eligible for inclusion if they enrolled participants aged 18 years and over of whom no less than 50% were undergoing major gastrointestinal surgery for cancer. Major gastrointestinal surgery for cancer was defined as surgery via any type of intra-peritoneal incision (e.g., open, laparoscopic, minimally invasive) to the abdomen, involving luminal resection or resection of a whole, or part of a, solid organ within the gastrointestinal tract and requiring general anaesthetic and a minimum overnight stay in hospital [6,12]. For this review, oesophagectomy was classified as gastrointestinal surgery.

#### 2.2.3. Interventions

Perioperative physiotherapy techniques/treatments with the intent of minimising the deleterious impacts of anaesthesia and major surgery [6,13] including but are not limited to PPCs, postoperative ileus, wound infection and deep vein thrombosis. Physiotherapy techniques include interventions delivered preoperatively to enhance a patient’s knowledge of their risk of a PPC, training on breathing exercises, and targeted respiratory muscle strengthening [5]. Postoperative interventions include lung expansion techniques, such as coached deep breathing exercises or incentive spirometry and targeted therapeutic physical activity [6]. For the purposes of this review, interventions had to be delivered with the intent of prophylaxis against postoperative complications not in response to signs of deterioration or a diagnosis of a complication.

Prehabilitation consisting of multi-modal or whole-body exercise in the gastrointestinal surgery population delivered with the aim of enhancing preparedness for surgery and reducing postoperative complications were not included in this systematic review as numerous systematic reviews summarising the evidence of this multimodal intervention on postoperative complications have been conducted [6]. Non-invasive ventilation (NIV) was also specifically excluded due to recent systematic reviews in the abdominal surgery population [14].

Interventions for this systematic review included the following:(i)Patient education and training to perform self-directed breathing exercises;(ii)Coached/supervised breathing exercises, without augmentation with a device;(iii)Respiratory muscle training;(iv)Positive expiratory pressure therapy;(v)Incentive spirometry;(vi)Postoperative supervised/assisted mobilisation and physical activity;(vii)Combinations of above.

The time point of therapy was recorded: preoperative only, postoperative only or perioperative. Eligible interventions were required to be delivered by physiotherapists (including exercise therapists, and respiratory therapists). This restriction was chosen as physiotherapists are the predominant profession providing these interventions in high-income countries. Physiotherapists are also specifically trained in the provision of these interventions with minimum competency standard thresholds and credentialing as part of entry level tertiary qualifications. This provides some quality assurance of the interventions delivered in included trials minimising the possible influence of unknown or low quality treatment fidelity on outcomes. Interventions delivered by a Therapy Assistant under explicit instruction from the physiotherapist were also included as this also is reflective of current practice in many hospitals.

#### 2.2.4. Comparator

A true no-treatment control (i.e., absence of any physiotherapy treatment) or usual care. Usual care in high-income country health care systems includes a culture of routine early mobilisation out of bed for patients post major surgery and the general encouragement of breathing exercise but not the provision of specific coaching for breathing exercises or the use of devices such as incentive spirometers [6,15]. Studies were excluded if the comparator involved participants receiving any type of active physiotherapy interventions beyond this definition.

#### 2.2.5. Outcomes

The primary outcome for this systematic review was the incidence of a PPC within the acute hospital stay including: composite PPC diagnostic tools [16], atelectasis as diagnosed from radiographic imaging or lung ultrasound, pneumonia (any diagnostic construct), acute respiratory failure (any diagnostic construct), or acute hypoxemia (arterial blood gases, pulse oximetry) [17]. Secondary outcomes are any type of reported in-hospital medical or surgical complication, functional recovery measures, length of hospital stay, 30 day mortality and health related quality of life. Measures of adherence to delivered interventions and any adverse events associated with the intervention will also be collected.

#### 2.2.6. Exclusion Criteria

A study was excluded if:-It was in a language other than English and a reliable translation could not be obtained.-More than 50% of participants had a surgical procedure for an indication other than cancer (i.e., bariatric surgery or cholecystectomy) or where the surgery did not involve the gastrointestinal tract (i.e., gynaecological or urological).-Participants had a diagnosis of a PPC on entry into trial.-It compared interventions to other active interventions, unless a sub-category comparing an active intervention to a suitable control could be extracted.-The intervention was part of a multimodal package of care and the independent effect of the physiotherapy intervention could not be determined, e.g., Enhanced Recovery After Surgery or fast-track pathways.-The intervention was prehabilitation of whole-body strength and/or conditioning-The intervention was NIV.

For further details of inclusion and exclusion criteria, see Appendix A.

### 2.3. Search Strategy

The search strategy was developed by the primary author (SW) in consultation with a medical librarian experienced in the development of health-related systematic review search strategies. Study investigators peer-reviewed the draft search strategy which then underwent piloting and validity refinement. The search string was developed for MEDLINE and modified appropriately for other databases (Appendix A, for full search strategy and MESH terms). No language or publication date restrictions were applied.

### 2.4. Information Sources

A systematic search was undertaken in MEDLINE via PubMed, Wed of Science, CINAHL Complete via EBSCO Host and COCHRANE Central on 21 and 22 August 2024. Reference lists of full-text articles selected for inclusion were screened for additional citations not captured in the original search strategy.

### 2.5. Study Selection

All identified records were imported into Covidence [18] where duplicates and ineligible studies were removed by automation tools then manually checked (SW). Two reviewers (SW and SM) independently conducted all stages of study review and selection. Search results were screened by title and abstract, with clearly ineligible articles excluded. All remaining articles underwent full-text review for eligibility against the inclusion criteria. A third party (IB) reviewed circumstances where consensus was not reached between the two reviewers, with their decision considered final. Study flow through the screening and selection process was captured in a PRISMA flow chart (see Figure 1). Explanations for article exclusion following full-text review are recorded (Appendix A).

### 2.6. Data Collection Process

Data were extracted from selected studies independently by two reviewers (SW and SM) using a standardised form. Disagreements on data extraction were addressed through a cooperative review of the original source, with any unresolved conflicts managed by a third party (IB). Where data in the published study were unclear, the primary author attempted to contact the corresponding author via email to request further information.

### 2.7. Data Items

Data extracted included the following:-Study details (author, year published, title, journal, country of recruitment).-Methods (design, randomization, blinding, number of study arms).-Participants (numbers recruited, demographics).-Intervention and comparator details.-For dichotomous outcome measures, the number of participants in each group meeting the outcome.-For continuous outcomes, the mean and variance for each group. When mean and variance were not reported in the original article, these were calculated from the raw data using Cochrane methods [19].

### 2.8. Risk of Bias

Three reviewers (SW, RM and JW) independently appraised each included study for methodological quality and risk of bias (RoB 2.0; www.riskofbias.info/ (accessed on 28 October 2024) [20]. Any disagreements were resolved by consensus, with referral if necessary to a fourth assessor (IB). The risk of bias scoring was not an eligibility criterion for this systematic review.

### 2.9. Data Synthesis

Extracted data were synthesised using descriptive statistics and, where possible, meta-analyses were conducted. Pooled data were analysed for the effect of physiotherapy compared to a control of no physiotherapy or usual care of early mobilisation alone on the rate of PPCs (the primary outcome). Where a study had more than one intervention arm and both arms complied with the inclusion criteria, data were amalgamated into a single intervention group for meta-analysis [19]. Sensitivity analyses were conducted on the primary outcome to determine (a) the effect of removing data from studies with a high risk of bias, and (b) the effect of physiotherapy on different PPC diagnostic constructs.

Sub-group analyses for the effect of physiotherapy intervention delivery timing (preoperative, postoperative, perioperative) on the primary outcome were performed. Where data were available, secondary analyses were performed for the effect of physiotherapy on all-cause surgical and medical complications and the acute hospital stay.

Where meta-analysis was conducted, data were pooled using commercially available software [21] with Mantel–Haenszel (MH) random-effects modelling and graphically represented. Heterogeneity of the true effect size within included studies was assessed using the 95% prediction interval (PI) rather than the I^2^ statistic [22]. The I^2^ statistic provides information about the proportion of variance in observed effects due to variance in the true effect rather than sampling error and does not provide information about how much effect sizes may vary between studies [22,23]. If the PI demonstrates that the effect size is consistent across studies, the impact of the intervention will be similar for all relevant populations. Should the PI demonstrate that the effect size varies substantially, inferences can be made as to whether the effect is always beneficial but with differing degrees of effect (trivial to moderate to large) or even that the effect may be beneficial in some instances and potentially not beneficial in others [23]. By employing the PI, insights can be gained into the clinical interpretation of heterogeneity by estimating the true treatment effects that can be expected in the treatment of future patients [24].

Dichotomous outcomes were calculated using risk ratios (RR) and 95% confidence intervals (95%CI). Continuous outcomes were calculated as mean differences (MD) when data were on a uniform scale and standardised mean differences (SMD) with 95%CI when data were on a different scale.

## 3. Results

### 3.1. Search Results

Systematic searches produced a total of 6555 records (Figure 1). After duplicate removal and screening of titles and abstracts, 263 records proceeded to full-text screening with 254 excluded. Of excluded studies, 69 were deemed ineligible as the surgery was not due to gastrointestinal cancer. An additional 39 studies were excluded because an active control was used as the comparator (i.e., incentive spirometry was compared to coached deep breathing exercises). Twenty studies were excluded as the intervention was delivered by a health professional other than a physiotherapist (see Appendix A, for detailed description of exclusion reasons). After exclusions, nine publications from eight RCTs met the inclusion criteria (Table 2) [25,26,27,28,29,30,31,32,33].

### 3.2. Study Characteristics

The nine included publications had sample sizes ranging from 41 to 441 participants. A total of 1418 participants were randomised into intervention or control groups. Loss-to-follow up was low, with primary outcome data available from 732 intervention group participants and 638 control group participants (combined 1370/1418 (97%)). Two studies involved a four-arm design where three different interventions were each compared with a no-treatment control [27,30]. The six other studies compared a single intervention arm with a control. There was one multicentre international study (Australian and New Zealand) reporting different outcomes over two publications [25,26]. The others were single-centre trials conducted in Sweden, Brazil, China, the Republic of Korea, India and the United Kingdom. Characteristics of the included studies can be seen in Table 2.

### 3.3. Risk of Bias

Methodological limitations and overall risk of bias was determined to be low for three studies [25,28,31] and moderate in three studies [29,30,32]. Two studies were assessed as high risk of bias [27,33], with possible outcome reporting limitations [27,33] and imbalanced randomization [33]. For example, 28% of controls had oesophageal cancer compared to 4% of intervention participants and 40% of the control group had stomach cancer compared to 24% of the intervention group in the study published by Singh et al. [33]. See Appendix A, for the risk of bias figure.

### 3.4. Participants

Participants were, on average, 62 years old (SD 16) and there was an equal representation of the sexes (52.5% (703/1338) male). One study restricted recruitment to only those at high risk of developing PPCs [28] whilst, conversely, one study excluded participants with pre-existing respiratory disease [33]. All other studies included all patients generally listed for gastrointestinal surgery [25,26,27,29,30,31,32] with a majority of participants having an American Society of Anesthesiologists (ASA) score of 1 or 2 (971/1338 (73%)). Seven studies included a variety of gastrointestinal surgery procedures [25,26,27,28,29,30,33], whilst two focused on colorectal surgery [31,32]. Participant characteristics for all included studies can be seen in Table 2.

### 3.5. Interventions

#### 3.5.1. Preoperative Alone

Three studies provided preoperative physiotherapy alone. Two provided a single education and training session (15–30 min) where patients were taught breathing exercises and instructed to start these independently on waking from their anaesthetic after surgery [25,29]. The other provided training in preoperative breathing exercises, augmented with either an incentive spirometer or respiratory muscle trainer, to be completed independently prior to surgery twice daily for a minimum of two weeks before surgery [27]. Refer to Table 3 for detailed intervention characteristics of the included studies that provided preoperative physiotherapy alone.

#### 3.5.2. Postoperative Alone

Three studies investigated postoperative physiotherapy. One provided intensive daily coached breathing exercise sessions for a minimum of ten sessions across the first four postoperative days [28]. Another provided three coached breathing exercise sessions per day with or without incentive spirometers for the first five postoperative days [30]. The final study provided twice-daily 15 min whole-body exercise sessions from the first postoperative day until hospital discharge [31]. Table 3 provides intervention characteristics for included studies that provided postoperative physiotherapy alone.

#### 3.5.3. Perioperative (Both Preoperative and Postoperative Interventions)

One study admitted patients to hospital five days before surgery where thrice daily education and coached breathing exercises and positive expiratory pressure blowing therapy was supervised with these therapies continuing for four days following surgery [32]. The final study provided outpatient preoperative supervised breathing exercises and incentive spirometry for 30 min twice daily for up to five days which continued postoperatively in hospital for an unspecified period [33]. Detailed intervention characteristics of the included studies that explored perioperative interventions can be found in Table 3.

### 3.6. Comparators

Four studies had a control group where participants were not provided with any physiotherapy [27,29,30,31]. In two studies, participants, including those in the control, were provided with a standardised postoperative ambulation protocol delivered by physiotherapists [25,26,28] whilst two studies provided a control of general encouragement of mobilisation but did not specify who provided this [32,33]. Table 3 provides details of the comparator for each included study.

### 3.7. Adherence to Intervention

Two studies reported participant adherence to physiotherapy interventions. One study recorded that the intervention group performed significantly more breathing and coughing exercises [28] and the other reported excellent adherence (85%) to prescribed physical exercise sessions [31]. Table 3 highlights adherence to interventions for each included study.

### 3.8. PPC Outcomes

Outcomes and the time point measured are found in Table 2. Two studies reported PPCs using composite diagnostic tools [25,28], whilst four reported PPCs as total incidence from a composite of different respiratory diagnoses [27,30,32,33]. The effect of physiotherapy on pneumonia was reported in four studies [25,29,32,33], two studies [25,32] reported acute respiratory failure, two reported hypoxaemia [26,29] and three reported atelectasis [26,32,33].

### 3.9. Effect of Physiotherapy Interventions on Primary Outcome

#### PPC

Seven of eight studies [25,27,28,29,30,32,33] measured the incidence of PPCs in a physiotherapy intervention group compared to no physiotherapy or early mobilisation alone and were included in the meta-analysis. The aggregated data from these existing studies suggest that exposure to a respiratory focused physiotherapy intervention is likely to result in an estimated 59% reduction in the risk of PPCs with 95% probability that the true effect is between a 27% to 77% reduction in risk (RR 0.41, 95%CI 0.23 to 0.73, *p* < 0.001). However, as existing studies vary significantly in effect direction, the calculated PI of 0.09 to 1.91 indicates there is a 95% probability that future studies in this field could be either positive or negative in effect (see Figure 2).

A sensitivity analysis was undertaken for the primary outcome excluding studies assessed at high risk of bias [27,33]. Removing these two studies from the analysis did not change the mean estimate of the effect of the intervention. However, the certainty of a clinically meaningful difference in PPC risk is lessened with the 95% confidence interval including the possibility of as little as a 3% reduction in PPC risk with exposure to a physiotherapy intervention (RR 0.48, 95%CI 0.24 to 0.97, *p* = 0.04, 95% PI 0.05 to 4.35).

Sensitivity analyses of the effects of physiotherapy on alternate PPC diagnostic constructs found a certain reduction in pneumonia risk (RR 0.37, 95%CI 0.21 to 0.63, *p* < 0.001), However, the effect of physiotherapy on acute respiratory failure (RR 0.40, 95%CI 0.15 to 1.07, *p* = 0.07), atelectasis (RR 0.52, 95%CI 0.24 to 1.11, *p* = 0.09) and acute hypoxemia (RR = 0.26, 95%CI 0.02 to4.36, *p* = 0.32) is less certain (Figure 3a).

A sub-group analysis was performed to explore if the timing of physiotherapy delivery influenced the primary outcome. Interventions delivered preoperatively (RR 0.44, 95%CI 0.29 to 0.67, *p* < 0.001) and perioperatively (RR 0.25, 95%CI 0.15 to 0.41, *p* < 0.001) may lead to a probable reduction in PPC risk, whereas interventions delivered only in the postoperative phase may be unlikely to effect PPC risk (RR 2.41, 95%CI 0.34 to 17.03, *p* = 0.38). See Figure 3b.

### 3.10. Effect of Physiotherapy Interventions on Secondary Outcomes

#### 3.10.1. All-Cause Complications

Due to significant heterogeneity amongst the classification and reporting of postoperative complications across the included studies, it was not possible to conduct a meta-analysis of the effect of physiotherapy on surgical complications separately to medical complications. For this review these outcomes were aggregated to all-cause complications. For specific details regarding the data extracted and aggregated from each included study for all-cause complications, please see Appendix A. Although the effect estimate favours physiotherapy interventions to reduce the risk of all-cause postoperative complications, the probability of a null effect has not been excluded (RR 0.87, 95%CI 0.71 to 1.05, *p* = 0.15, Figure 4).

#### 3.10.2. Hospital Length of Stay

Pooled data from seven studies reporting hospital stay [25,28,29,30,31,32,33] suggest an estimated reduction in a day less in hospital for participants provided with physiotherapy (MD–1.4 days, 95%CI −2.24 to −0.58, *p* = 0.01, Figure 5). Sub-group analyses according to timing of the physiotherapy intervention finds that the extent of this benefit may be influenced by the timing of physiotherapy delivery, with perioperative physiotherapy indicating a more beneficial effect on length of stay than postoperative physiotherapy alone.

#### 3.10.3. Functional Outcomes

Due to significant heterogeneity in the assessment of functional outcomes, a meta-analysis was not possible. Of the three studies that reported functional outcomes, none reported a significant difference between groups [25,28,31].

#### 3.10.4. All-Cause Mortality

Pooled data from three studies reporting in-hospital all-cause mortality [25,28,30] did not detect an effect favouring physiotherapy (RR 1.01, 95%CI 0.30 to 3.40).

#### 3.10.5. Quality of Life

No study reported HRQoL within the first 30 days post-surgery.

#### 3.10.6. Adverse Events

There were no adverse events reported to be directly associated with physiotherapy interventions [25,31].

## 4. Discussion

This systematic review and meta-analysis aggregated data from eight RCTs that investigated physiotherapy interventions to prevent postoperative complications and improve recovery following major gastrointestinal surgery for cancer. This is the first systematic review to consider the effect of physiotherapy compared to providing no physiotherapy or postoperative early mobilisation alone in this specific cancer surgery population. The meta-analysis of pooled data finds that physiotherapy is likely to be effective in halving the risk of a patient developing a PPC, including pneumonia, and in reducing acute hospital length of stay by one to two days. These data also suggest that these effects are most likely to be achieved if physiotherapy is delivered in both the pre- and postoperative phases of gastrointestinal surgery, with the benefits of a postoperative-alone physiotherapy service uncertain.

This new meta-analysis aligns with earlier systematic reviews exploring physiotherapy for the prevention of PPCs in patients undergoing abdominal surgery. Two previous meta-analyses concluded that prophylactic respiratory physiotherapy is likely to be effective in reducing PPCs [7,9]. Two other meta-analyses that report greater uncertainty about the effect of physiotherapy may be limited as both specifically excluded preoperative physiotherapy interventions [8,10].

High-quality RCTs assessing the effectiveness of preoperative education and deep breathing exercise training in this patient population have consistently reported positive effects [25,27,34,35]. At present, the evidence appears to support the provision of preoperative interventions including physiotherapy education and training [34], inspiratory muscle training [36], and prehabilitation as the most effective way of preventing PPCs [37]. Data from our study support these previous studies, suggesting that the timing of prophylactic interventions influences the risk of PPCs, with preoperative physiotherapy likely to reduce the risk of PPCs after gastrointestinal surgery for cancer and with the effects of postoperative physiotherapy alone uncertain. This may directly challenge traditional paradigms of care where hospitals in high-income countries predominantly only provide physiotherapy postoperatively [38,39,40].

From a clinical perspective, postoperative physiotherapy alone may not be sufficient to prevent postoperative complications in this patient population and preoperative physiotherapy appears to have greater effect in reductions in complications such as PPCs. However, the limited number of studies investigating physiotherapy for patients having major gastrointestinal surgery for cancer and the inconsistent effect direction mean that currently, there are not enough data to provide firm recommendations for practice change in this area. A meta-analysis with greater scope is required before any practice recommendations can be made for postoperative physiotherapy, and therefore, the authors currently advise that clinicians maintain the status quo of current physiotherapy service delivery with regard to postoperative physiotherapy.

The initial intent of this review was to analyse a broad range of physiotherapy interventions for patients having gastrointestinal surgery for cancer; however, following the application of stringent predetermined inclusion criteria, only nine publications from eight RCTs were eligible for inclusion. There were several considerations guiding the application of stringent inclusion criteria for this review. Most abdominal surgery RCTs define patient populations via incision type or location [6] rather than by indication for surgery. Consequently, 69 RCTs were excluded because it could not be established that at least 50% of patients had specifically undergone gastrointestinal surgery for cancer. Studies of multimodal Enhanced Recovery After Surgery (ERAS) programmes aimed at reducing an array of postoperative complications and enhancing outcomes [15,41] added complexity to the screening process. Due to the multi-modal, multi-professional nature of the ERAS interventions, effects of the physiotherapy intervention could not be independently determined, deeming these studies ineligible.

Specifying the comparator to be a true no-treatment control or clearly defined usual care of early mobilisation further restricted study eligibility with 39 clinical trials excluded on this basis. Many RCTs exploring the efficacy of physiotherapy interventions employ a direct comparison of one active intervention compared to one or more other active interventions. Whilst this methodology resolves how effective one intervention is compared to another, it does not resolve whether an intervention provides any additional benefit compared to no treatment or standardised usual care [19]. Given the current uncertainty regarding whether additional physiotherapy beyond usual care confers greater benefits in the broader abdominal surgery population, it was necessary to include only studies that met the comparator eligibility criteria.

Only interventions specifically delivered by physiotherapists (or equivalent) were included in this review, resulting in a further 20 studies being excluded. Although respiratory exercises and postoperative therapeutic rehabilitation may be delivered by health professionals other than physiotherapists (or equivalent), these interventions are commonly considered a primary defined role of physiotherapists working within the multidisciplinary hospital teams caring for this patient population in high-income countries. Furthermore, physiotherapists undergo specific training in the provision of these interventions as part of minimum core competency thresholds to achieve entry level tertiary qualifications. The quality and standardisation of respiratory and rehabilitation interventions delivered by other health professions is uncertain. To ensure that the results of this systematic review were unlikely to be influenced by questions of treatment fidelity and quality of the interventions, studies were restricted to only those where the interventions were provided or prescribed by qualified registered physiotherapists.

In the studies included in this current review, reporting of interventions and outcomes lacked clarity and consistency, at times affecting the data that could be extracted for analysis. Only two studies reported on adverse events attributed to physiotherapy intervention as a possible outcome [25,26,31]. Future trials could be improved by clearer detailed reporting of intervention dosage and patient adherence and this may allow better insights into feasibility of interventions for clinical practice and acceptability of interventions to clients [42]. It is recommended that all future RCTs prospectively collect and report on data regarding both the benefits and harms of health interventions [43]. Heterogenous outcome measures, especially for functional outcomes, impacted how findings were reported and whether these findings could be combined for meta-analysis. The use of standardised outcome measures in perioperative medicine is required to facilitate more accurate meta-analyses and reduce research waste [44,45]. Future RCTs should carefully consider these factors when selecting and reporting on appropriate outcomes measures.

The findings of this current meta-analysis may be limited by the small number of included studies and, consequentially, uncertainty in effect size estimates restricting the ability to make specific practice-related recommendations. Unanswered questions remain in this patient population including uncertainty regarding efficacy of postoperative mobilisation and efficacy of coached postoperative breathing exercises. High quality RCTs with appropriate comparators are required to advance understanding in this area of practice. Additional research is necessary to further validate the observed effects of preoperative physiotherapy, to strengthen confidence in making change of practice recommendations. Employing broader eligibility criteria by including clinical trials testing respiratory and whole-body exercise interventions delivered by any type of health care provider and in participants having all-cause abdominal surgery, not just for cancer, would have resulted in an additional 89 studies being included for data pooling. This additional data may have allowed for more definitive conclusions.

Future reviews in this field would be advised to expand eligibility criteria as outlined, particularly regarding interventions delivered by any health care provider and utilise sensitivity analyses to elicit whether intervention efficacy is influenced by the health discipline who provides that care. Moving forwards, a review of broader scope and with component network analysis [46] will assist with addressing remaining questions in this field of practice.

## 5. Conclusions

This systematic review contributes to previous findings supporting the hypothesis that prophylactic physiotherapy interventions may minimise the risk of PPCs after major surgery. This current review finds that the size of this effect specifically for patients having gastrointestinal surgery for cancer could be a large and significant reduction in risk of between 27% and 77% compared to patients who do not receive any type of physiotherapy or early mobilisation alone. These findings should be interpreted with caution given the low number of eligible studies and the heterogeneity of the included studies. The strength of the existing body of literature regarding preoperative physiotherapy indicates that preoperative physiotherapy should be implemented in this patient population. However, completion of a broader network meta-analysis is essential prior to considering any recommendations for practice changes in the delivery of postoperative physiotherapy. Until further evidence is available, clinicians who currently provide postoperative physiotherapy to patients undergoing major gastrointestinal cancer surgery are advised to maintain the status quo of their service delivery.

## Figures and Tables

**Figure 1 cancers-17-00676-f001:**
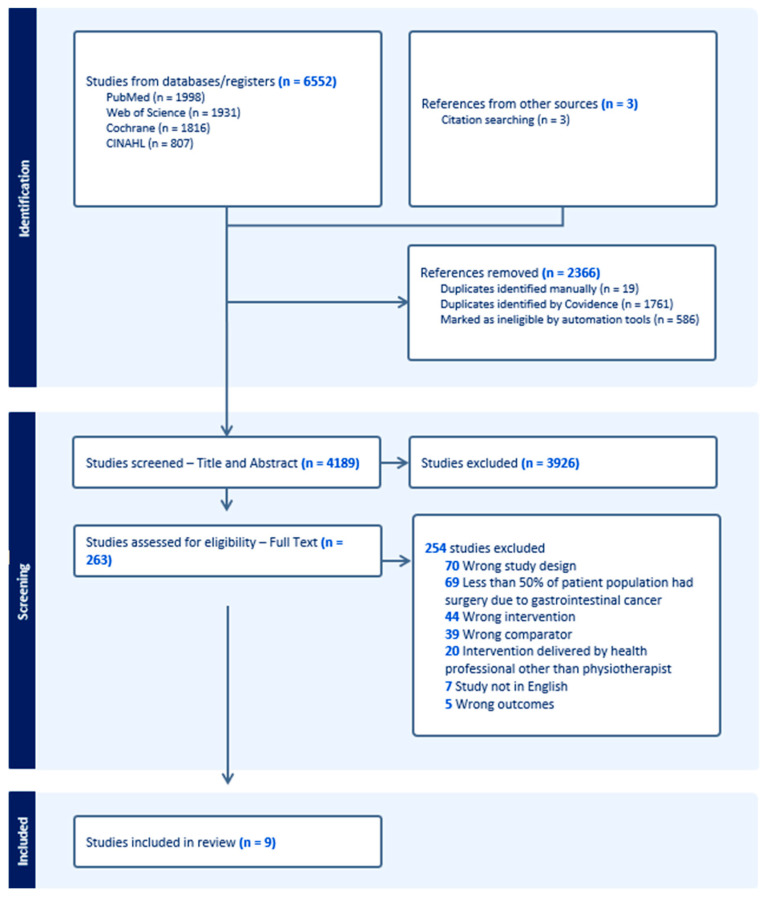
PRISMA flow chart of included and excluded studies within this systematic review and meta-analysis.

**Figure 2 cancers-17-00676-f002:**
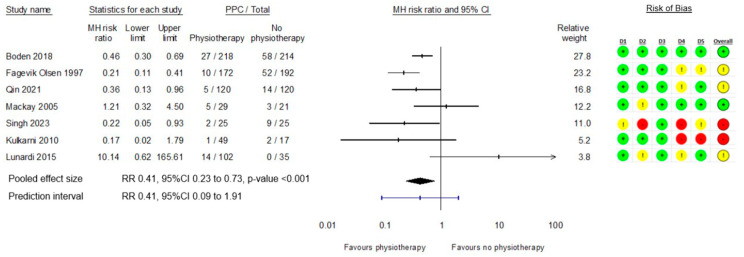
Meta-analysis of effect of physiotherapy on PPCs with risk of bias grading [25,27,28,29,30,32,33].

**Figure 3 cancers-17-00676-f003:**
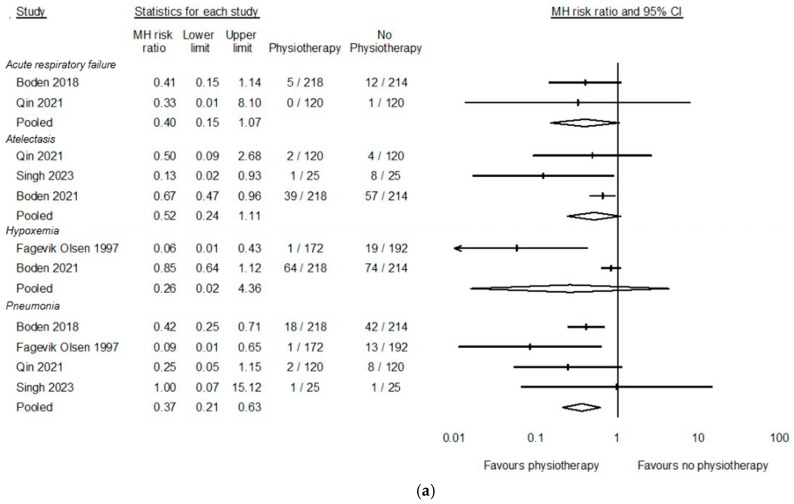
(**a**). Sensitivity analysis of effect of physiotherapy on different PPC diagnostic constructs [25,26,29,32,33]; (**b**). Sub-group analysis of the effect of time of delivery of physiotherapy on PPCs [25,27,28,29,30,32,33].

**Figure 4 cancers-17-00676-f004:**
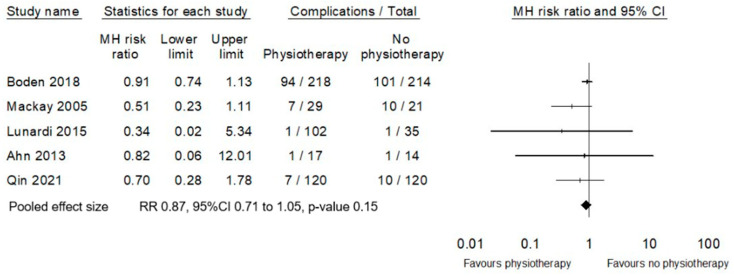
Meta-analysis of effect of physiotherapy on all-cause complications (excluding PPC) [25,28,30,31,32].

**Figure 5 cancers-17-00676-f005:**
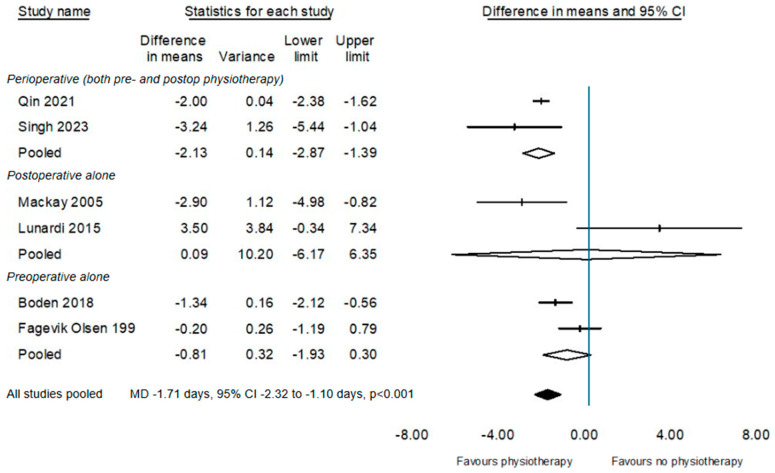
Sub-group of the effect of physiotherapy on acute hospital length of stay according to timing of delivery of physiotherapy intervention [25,28,29,30,32,33].

**Table 1 cancers-17-00676-t001:** Study inclusion criteria.

Criteria	Category	Description
Inclusion criteria	Design	RCTs, pseudo RCTs or cluster RCTs
	Participants	Adults undergoing gastrointestinal surgery for cancer
	Intervention	Perioperative physiotherapy interventions provided with the intent of minimising adverse effects of anaesthesia and surgery
	Comparison	True no treatment control or the usual care of early mobilisation alone
	Outcomes	Primary outcome—PPCs. Secondary outcomes—adherence to interventions; in-hospital medical and surgical complications, adverse events, and functional status; length of acute hospital stay; 30-day mortality and health related quality of life

Legend: RCTs—randomised controlled trials, PPCs—postoperative pulmonary complications.

**Table 2 cancers-17-00676-t002:** Characteristics of included studies.

Study; Country	Total Participants Randomised and Reported	Participant Characteristics			Intervention Timing	Outcomes	Time Point of Primary Outcome Measurement
		Age (Years)	Gender (Male, n/N, %)	ASA (Grade, n/N, %)			
Ahn, 2013 [31]; Korea	Total randomised: 41 Data reported: I: 17 C: 14	All: NR I: 56 (SD 7) C: 57 (SD 6)	17/31 (55%)	1–2, 31/31 (100%)	Postoperative	Primary: Acute hospital LOS Secondary: Time to flatus, functional outcomes, all-cause complications, readmission within 30 days of discharge	Daily from POD 1—discharge
Boden, 2018 [25] and, 2021 [26]; Australia and New Zealand	Total randomised: 441 Data reported: I: 218 C: 214	All: NR I: 62 (SD 15) C: 66 (SD 14)	266/432 (62%)	1–2, 274/432 (64%) 3–4 157/432 (36%)	Preoperative	Primary: PPC Secondary: Pneumonia, atelectasis, hypoxaemia, readiness for hospital discharge, ICU/HDU LOS, ICU/HDU unplanned admissions, acute hospital LOS, all-cause postoperative complications, hospital costs, self-reported HRQOL and physical function, 12-month all-cause mortality.	Daily from POD 1–POD 7. POD 7–POD 14 as clinically indicated
Fagevik Olsen, 1997 [29]; Sweden	Total randomised: 368 Data reported I: 172 C: 192	All: 53 I: NR C: NR	158/368 (43%)	1–2, 313/368 (85%) 3–4, 48/368 (13%) ^	Preoperative	Primary: PPC Secondary: Pneumonia, hypoxaemia, FVC, PEFR, CXR, prophylactic antibiotics, ICU and hospital LOS	POD 1, POD 3, POD 6
Kulkarni, 2010 [27]; United Kingdom	Total randomised: 80 Data reported: I 1: 17 I 2: 15 I 3: 17 C: 17	All: NR I: NR C: NR	NR	NR	Preoperative	Primary: MIP, MEP, VC, FVC, FEV1 Secondary: PPC, respiratory rate, oxygen saturation on room air, HDU/ICU LOS, acute hospital LOS,	48 h pre-surgery, once between POD 1- POD7
Lunardi, 2015 [30]; Brazil	Total randomised: 137 Data reported: I 1: 33 I 2: 35 I 3: 34 C: 35	All: NR I 1: 63 (SD 14) I 2: 58 (SD 14) I 3: 55 (SD 12) C: 57 (SD 17)	59/137 (43%)	1–2, 137/137 (100%)	Postoperative	Primary: PPC, Thoracoabdominal mechanics via optoelectronic plethysmography Secondary: Atelectasis, hypoxaemia when SpO_2_ < 85%, pneumonia, spirometry, pain score (VAS)	24 h pre-surgery, POD 3
Mackay, 2005 [28]; Australia	Total randomised: 56 Data reported: I: 29 C: 21	All: 66 (SD 14) I: 63 (SD 13) C: 69 (SD 15)	25/50 (50%)	1–2, 28/50 (56%) 3–4, 22/50 (44%)	Postoperative	Primary: PPC Secondary: Supplemental oxygen requirement, postoperative mobility, physiotherapist time, total number of physiotherapy treatments	Daily from POD1
Qin, 2021 [32]; China	Total randomised: 245 Data reported: I: 120 C: 120	All: NR Intervention: 63 (SD 13) Control: 62 (SD 12)	146/240 (61%)	1–2, 151/240 (63%) 3–4, 89/240 (37%)	Perioperative	Primary: PPC Secondary: Pneumonia, atelectasis, respiratory failure, arterial oxygenation, all- cause postoperative complications, LOS, hospital costs, patient satisfaction mortality	POD 1, POD 4
Singh, 2023 [33]; India	Total randomised: 50 Data reported: I: 25 C: 25	All: 53.5 (SD 12.92) Intervention: NR Control: NR	32/50 (64%)	1–2, 37/50 (74%) 3, 13/50 (26%) (excluded ASA > 3)	Perioperative	Primary: PPC Secondary: Pneumonia, atelectasis, FEV1, FVC, ICU LOS, hospital LOS, length of surgery	Day of surgery, POD 1, POD 3, POD 7, POD 15, POD 30

Legend: ASA = American Society Anesthesiologists score, where 1 is a normal healthy patient, 2 is a patient with mild systemic disease, 3 is a patient with severe systemic disease, 4 is a patient with severe systemic disease that is a constant threat to life, and 5 is a moribund patient who is not expected to survive. FEV1 = force expiratory volume in one second; FVC = forced vital capacity; HRQOL = health related quality of life; ICU or HDU = Intensive care unit or High dependency unit; LOS = length of stay (days); MEP = maximal expiratory pressure; MIP = maximal inspiratory pressure; PEFR = peak expiratory flow rate; POD = postoperative day; PPC = postoperative pulmonary complication; VC = vital capacity; I = Intervention; C = Control; ^ = missing ASA data for 7 participants; n/N = number of participants with characteristic/total number in cohort; NR = not reported.

**Table 3 cancers-17-00676-t003:** Intervention characteristics of included studies.

Study (Country)	Control	Intervention	Setting (Delivery)	Intervention Dosage	Frequency	Adherence to Intervention	Adverse Events
Ahn, 2013 [31] Korea	Unsupervised sitting and walking postoperatively. No respiratory physiotherapy	In addition to the control therapies patients were provided with supervised whole-body exercise. No respiratory physiotherapy.	Postoperative in-hospital. Face to face.	15 min per session	Twice daily from POD 1 until discharge from acute care	Adherence to intervention of 84.5%	Collected as an outcome. No adverse events reported.
Boden, 2018 [25] and 2021, [26] Australia and New Zealand	Preoperative assessment with a physiotherapist and an information booklet provided. Postoperatively a standardised mobilisation protocol was delivered by physiotherapists to all participants. Brief postoperative check to confirm performing breathing exercises as in booklet	In addition to the control therapies patients were provided with preoperative education and training to perform self-directed breathing exercises and coughing (DB&C) exercises	Preoperative in preadmission clinic. No more than 6 weeks prior to surgery	30 min	Single session	Not recorded	Collected as an outcome. No adverse events reported.
Fagevik, Olsen 1997 [29] Sweden	No preoperative education or training. No postoperative physiotherapy.	Patient education and training to perform self-directed DB&C exercises, addition of PEP therapy for high risk. Postoperative review to confirm performing DB&C exercises as taught preoperatively	Preoperative in-hospital day prior to surgery. Face to face.	Preoperative: 10–15 min. Postoperative: 15–20 min	Two sessions	Not recorded	Not collected as an outcome. Unknown.
Kulkarni, 2010 [27] United Kingdom	Preoperative assessment. No preoperative education or training	Preoperative breathing exercise training. 3 intervention arms—DB&Cexercises without devices; incentive spirometry; inspiratory muscle training	Preoperative outpatient clinic at least 2 weeks prior to surgery. Face to face education and training. Independent exercise following the initial education and training session	Unclear duration of education session. Independent training sessions 15 min per session	Single session of preoperative education and training. Twice daily respiratory training sessions for a minimum of 2 weeks.	Not recorded	Not collected as an outcome. Unknown.
Lunardi, 2015 [30] Brazil	No intervention	Postoperative DB&C exercise training. 3 intervention arms—DB&Cexercises without devices; flow incentive spirometry; volume incentive spirometry	Postoperative in-hospital	5 sets of 10 repetitions	Three times daily from POD 1 to POD 5	Not recorded	Not collected as an outcome. Unknown.
Mackay, 2005 [28] Australia	standardised postoperative mobilisation delivered by physiotherapists	In addition to directed early mobilisation, postoperative DB&C exercises with a device	Postoperative in hospital	2 sets of 3 deep breaths followed by forced exhalation.	Three times daily on POD 1 and 2. Two times daily on POD 3 and 4. Once daily if required thereafter	Intervention group performed significantly more DB&C exercises; 99 repetitions (95%CI 82 to 116)	Not collected as an outcome. Unknown.
Qin, 2021 [32] China	Usual perioperative care, general encouragement of mobilisation	In addition to the control therapies, patients were provided with pre- and postoperative education and coached DB&C exercises with PEP therapy.	In-hospital pre- and postoperatively.	5 repetitions of deep breath and cough, 5 repetitions of PEP, pursed lip breathing 10 min	3 sessions daily from preoperative day 5 until POD 4	Compliance with exercise sessions. Data not provided.	Not collected as an outcome. Unknown.
Singh, 2023 [33] India	Standard/usual perioperative care	Preoperative education and DB&C exercise training, plus incentive spirometry. Continued postoperatively.	Preoperative outpatient clinic from at least 2 days preoperatively. Postoperatively in hospital setting	30 min sessions	2 sessions daily	Not recorded	Not collected as an outcome. Unknown.

Legend: DB&C = deep breathing and coughing; PEP = positive expiratory pressure; POD = postoperative day.

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
