# Peer review of "Interventions Provided by Physiotherapists to Prevent Complications After Major Gastrointestinal Cancer Surgery: A Systematic Review and Meta-Analysis"

_cancers, 2025, doi:10.3390/cancers17040676_

Round 1

Reviewer 1 Report

Comments and Suggestions for Authors

The manuscript presented for review is an interesting analysis of the importance of physiotherapy interventions in preventing and pulmonary complications after major gastro-intestinal surgery. The authors did this by performing a systematic review and meta-analysis of existing data. 

The methodology is well described and follows a PROSPERO registered protocol. The search strategy, inclusion and exclusion criteria, and data collection are well described and are sufficient to allow reproductibility of study.

My concerns and suggestions are as follows:

1. Please reformat the layout of Table 2. - landscape would improve radability

2. Please use single spacing between all lines of text

3. Figures 3a and 5 need reformating (3a low resolution, 5 bad aligment)

4. Please discuss the influence of physiotherapy interventions in preventing other complications associated with major gastro-intestinal surgery (such as thromboembolic events or septic complications) or rename the paper to better reflect the fact that only lung complications are discussed.

5. Please add an explanation for why the 20 studies in which the physiotherapy interventions  were delivered by other types of healthcare professionals were excluded. Given the large number of studies, even with an expected lower efficacity of the intervention, they could still influence the overall results - high risk of bias and this should be addressed and perhaps admited as a limitation of the study.

6. The analysis should include if possible details about the type of surgery, stage of cancer, biological particularities of the patients (for instance malnutrition) - other factors may influence the postop complications and induce a risk of bias. Please add such discussions. 

Author Response

Dear Reviewer,

Please see the attachment for responses to reviewer comments.

Kind regards.

Reviewer 2 Report

Comments and Suggestions for Authors

Congratulations on the article. I have no negative remarks to make. I think there needs to be a standardized protocol for procedures.

Author Response

Dear Reviewer,

Please see attachment for response to reviewer comments.

Kind regards.

Reviewer 3 Report

Comments and Suggestions for Authors

1. Introduction:

The introduction considers incorporating more recent studies or statistics reflecting the prevalence of complications after gastrointestinal cancer surgeries.

Briefly articulating the gap in the existing literature that this review addresses would strengthen the rationale for the investigation.

2. Methods:

The methods section might include a comprehensive overview of the literature search strategy, including databases searched, keywords utilized, and the date range of studies considered.

A clear rationale for the selection criteria (inclusion/exclusion) for studies reviewed would provide transparency to the research process.

The application of statistical methods during meta-analysis should be explicitly described, ensuring reproducibility of results.

3. Results:

Results need to be presented clearly, ensuring all statistical analyses are explained in a manner accessible to the reader.

Visual aids such as tables and figures should be referred to explicitly within the text to guide the reader through the findings.

Discussing the significance of results comprehensively would provide better insight into the implications for clinical practice.

4. Discussion:

Discuss the limitations of the included studies and the potential impact these limitations may have on the overall findings of this review. Addressing biases (e.g., publication bias, reporting bias) is crucial for the conclusions' validity.

The implications of these findings should extend beyond the literature, providing recommendations for practice and future research avenues.

The conclusion summarizes the findings well but lacks emphasis on the implications for physiotherapy practice. Consider also including a call for further research exploring the timing and specific protocols of physiotherapeutic interventions.

Author Response

(The authors gave the same response as above.)

Round 2

Reviewer 1 Report

Comments and Suggestions for Authors

The last paragraph added to conclusions needs a bit of syntax adjustment "perioperative literature".

Otherwise the authors have addressed all my concerns adequately and the manuscript can be published in the current form after correcting the last paragraph. 

Author Response

Dear Reviewer,

Please find responses in the attached document.

Kind regards

Reviewer 3 Report

Comments and Suggestions for Authors

The Author responded to the comments.

Author Response

Dear Reviewer,

Please find the responses in the attached document.

Kind regards
